# Vitamin D Signaling in Psoriasis: Pathogenesis and Therapy

**DOI:** 10.3390/ijms23158575

**Published:** 2022-08-02

**Authors:** Anna A. Brożyna, Radomir M. Slominski, Bogusław Nedoszytko, Michal A. Zmijewski, Andrzej T. Slominski

**Affiliations:** 1Department of Human Biology, Institute of Biology, Faculty of Biological and Veterinary Sciences, Nicolaus Copernicus University, 87-100 Toruń, Poland; 2Department of Genetics, University of Alabama at Birmingham, Birmingham, AL 35294, USA; rslominski@uabmc.edu; 3Informatics Institute, University of Alabama at Birmingham, Birmingham, AL 35294, USA; 4Department of Dermatology, Allergology and Venerology, Medical University of Gdańsk, 80-211 Gdańsk, Poland; boguslaw.nedoszytko@gumed.edu.pl; 5Cytogeneticr Laboratory, Invicta Fertility and Reproductive Centre, 80-850 Gdańsk, Poland; 6Department of Histology, Medical University of Gdańsk, 80-211 Gdańsk, Poland; mzmijewski@gumed.edu.pl; 7Department of Dermatology, University of Alabama at Birmingham, Birmingham, AL 35294, USA; 8Laboratory Service, VA Medical Center at Birmingham, Birmingham, AL 35233, USA

**Keywords:** psoriasis, vitamin D, CYP11A, VDR, RORα, RORγ, megalin

## Abstract

Psoriasis is a systemic, chronic, immune-mediated disease that affects approximately 2–3% of the world’s population. The etiology and pathophysiology of psoriasis are still unknown, but the activation of the adaptive immune system with the main role of T-cells is key in psoriasis pathogenesis. The modulation of the local neuroendocrine system with the downregulation of pro-inflammatory and the upregulation of anti-inflammatory messengers represent a promising adjuvant treatment in psoriasis therapies. Vitamin D receptors and vitamin D-mediated signaling pathways function in the skin and are essential in maintaining the skin homeostasis. The active forms of vitamin D act as powerful immunomodulators of clinical response in psoriatic patients and represent the effective and safe adjuvant treatments for psoriasis, even when high doses of vitamin D are administered. The phototherapy of psoriasis, especially UVB-based, changes the serum level of 25(OH)D, but the correlation of 25(OH)D changes and psoriasis improvement need more clinical trials, since contradictory data have been published. Vitamin D derivatives can improve the efficacy of psoriasis phototherapy without inducing adverse side effects. The anti-psoriatic treatment could include non-calcemic CYP11A1-derived vitamin D hydroxyderivatives that would act on the VDR or as inverse agonists on RORs or activate alternative nuclear receptors including AhR and LXRs. In conclusion, vitamin D signaling can play an important role in the natural history of psoriasis. Selective targeting of proper nuclear receptors could represent potential treatment options in psoriasis.

## 1. Psoriasis: An Overview of the Clinical Problem

Psoriasis is a systemic, chronic, immune-mediated disease that is characterized by raised patches on the skin that affects approximately 2–3% of the world’s population [1]. The most common type of psoriasis is plaque psoriasis, which accounts for about 80–90% of cases. The other types include pustular psoriasis, which is more common in adults; guttate psoriasis, which is common in children; inverse psoriasis; and erythrodermic psoriasis. Psoriatic lesions are usually found on the scalp, skin folds, hands, feet, nails, and genitals [1]. Psoriasis usually manifests with cutaneous symptoms such as red, dry skin with raised, inflamed patches, silver scales or plaques, itch, thick, pitted nails, and swelling [1,2]. The psoriatic plaques are formed as an effect of epidermal hyperplasia resulting from enhanced proliferation and disturbed differentiation of keratinocytes [3]. These manifestations are related to the inflammatory process since psoriasis is an immune-mediated disease that is caused by the dysfunction of the immune system that resulted in inflammation [4,5,6]. The etiology and pathophysiology of psoriasis are still unknown, but the activation of the adaptive immune system with the main role of T-cells is key in psoriasis pathogenesis [2]. It is suggested that the impaired balance between T helper Type 1 (Th1) and Type 2 (Th2) cells, as well as cytokine production are the top causative factors of psoriasis [2,4,7,8,9]. In psoriatic patients, there is a shift towards the Th1 phenotype, which is characterized by the increased expression of IL-2, IFN-gamma, IL-12, T-bet [7,8], and the attenuation of the Th2 phenotype, with decreased expression of GATA3 and IL-4 [8] was found. In addition, the increased expression of IL-23 resulted in the increased level of Th17 and Th22 lymphocytes and their cytokines (IL-6, IL-20, IL-17, and IL-22) [2,9,10,11]. The production of Th17 cytokines in psoriasis is also related to the impairment function of regulatory T-cells (Tregs) [9,12,13,14,15]. Apart from T-cells, the other cells that are linked to psoriasis pathogenesis are the following: innate lymphoid cells, dendritic cells, mast cells, monocytes and macrophages, neutrophils, natural killers, keratinocytes, and many others (reviewed in [3,16,17]. Some data indicate dendritic cells activation that is mediated by peptide LL-37 and self-DNA, resulting in interferon production as a trigger for psoriasis pathogenesis [18,19]. Dendritic cells also promote the Th1 phenotype and the production of Th1 cytokines [19]. Figure 1 presents the major effector cells and signaling pathways in the immunopathogenesis of psoriasis.

The immunopathogenesis of psoriasis involves a complex inflammatory cascade, which is initially triggered by innate immune cells (keratinocytes, dendritic cells, NKT cells, macrophages, fibroblasts, γδ T-cells) that are activated by external (trauma, UV, microorganisms, drugs, smoking, diet and obesity, etc.) or internal factors (stress, autoantigens, DNA/RNA AMP complex, etc.) in genetically predisposed individuals. Cytokines that are produced by innate cells activate myeloid dendritic cells to increase the production of cytokines that are involved in the differentiation of lymphocytes to main adaptive immune cells: Th1, Th22, and Th17 which play the central role in the disease pathogenesis. Cytokines that are produced by these cells, which include: TNFα, IL-22, and IL-17A/F lead to keratinocyte proliferation, neoangiogenesis, chemokines production, neutrophils and CD8^+^ cells migration to the epidermis, and chronic inflammatory process. For this reason, biologic drugs targeting ILs such as IL 17, 23, and TNFα are the mainstay in the management of severe psoriasis [20,21]

## 2. Psoriasis and the Local Endocrine Regulators

The skin acts as a barrier and the organ that is involved in the coordination of the responses to environmental stimuli producing neurotransmitters, neuropeptides, and hormones [22], including the hypothalamic-pituitary-adrenal (HPA) axis [23,24,25,26]. The disturbances in steroidogenesis and the feedback of cutaneous neurohormones contributes to inflammation, as well as psoriasis development [27,28,29]. The corticotropin-releasing hormone (CRH) is overexpressed in areas of inflammation, as well as in peripheral tissues including the skin [30], thus its dysregulation is an important factor in psoriasis pathogenesis [31,32,33,34,35]. Correspondingly, aberrant CRH-receptor 1 is linked to psoriasis [28,36,37,38]. The increase of CRH in psoriasis can be accompanied by the up-regulation of proopiomelanocortin (POMC) [39], melanin-concentrating hormone receptor (MCHR1), and melanocortin receptors 2, 3 and 4 [40].

The glucocorticoids maintain skin barrier integrity and controls the inflammatory response by inhibiting the expression of IL-4 and IL-5 [41,42,43]. Glucocorticoid receptor level is decreased, and its activity is impaired in psoriasis [43]. The steroidogenic acute regulatory protein (StAR) that is expressed among other by keratinocytes [44,45], is involved in steroidogenesis and acute steroidogenic response [46]. In psoriasis, the expression of StAR is lowered or undetected [47,48], underlying the role of glucocorticoids in inflammatory skin diseases. In psoriasis several enzymes that are active in steroid synthesis are decreased: CYP11A1 (involved in cholesterol to pregnenolone conversion), CYP17 (involved in pregnenolone to cortisol conversion), 11βHSD1 (activator and inactivator of glucocorticoids), and 11βHSD2 (inactivator of cortisol by converting to its ketone form) [43,45,49]. It must be noted that CYP11A1 is expressed in the skin [44] and immune cells [36] and cortisol and corticosterone are produced by skin cells under tight regulatory mechanisms [26,50,51,52]. The deregulation of this process can lead to inflammatory diseases including psoriasis [28,33].

Since inflammation is the main process in the pathogenesis and development of psoriasis, it is recognized that other immune-mediated inflammatory diseases are common psoriasis comorbidities, e.g., psoriatic arthritis, cardiovascular disease, metabolic syndrome, inflammatory bowel diseases [2,4]. It should be noted that psoriatic patients are more likely to develop severe vascular events such as myocardial infarction and stroke (up to 50%) than the general population [53,54]. In addition, among psoriatic patients, different endotypes with different risks, severity, and treatment options are distinguished based on hypertension, red cell distribution width, and the mean platelet volume [55,56]. Thus, the modulation of the local neuroendocrine system with the downregulation of pro-inflammatory and upregulation of anti-inflammatory messengers represent a promising adjuvant treatment in psoriasis therapies [28].

## 3. Vitamin D Endocrine System

### Classical and Non-Classical Activation Pathways

The exposure of keratinocytes to ultraviolet B radiation initiates the 7-dehydrocholesterol (7DHC) photochemical transformation to a pro-hormone vitamin D_3_ [57,58,59]. Its transformation activation requires two-steps: hydroxylations at C25 (by CYP2R1 and CYP27A1) and C1α (by CYP27B1) to produce 1,25(OH)_2_D_3_. The cutaneous synthesis supplies more than 90% of its body’s requirement [57,59,60]. 1,25(OH)_2_D_3_, in addition to regulating calcium homeostasis, has important pleiotropic effects affecting almost all body functions. This action is mediated through interactions with vitamin D receptor (VDR), belonging to a subfamily of nuclear receptors [57,58,59,61,62,63]. VDR heterodimerizes with the retinoid X receptor (RXR) and functions as a ligand-activated transcription factor, after binding to the promoter regions of VDR responsive element (VDRE) to influence the expression of responsive genes [58,63,64]. Not only the expression of the VDR receptor determines the responsiveness of the cells to vitamin D, but also its polymorphisms. It was shown that the F and T alleles of Fok1 and Taq1 have been associated with increased VDR activity [65]. There is growing evidence that induction of transcriptional activity of VDR by 1,25(OH)_2_D_3_ does not fully explain the complexity and variety of cellular responses to this multipotent hormone. Thus, so called alternative, non-genomic response has been described. It was suggested that this rapid response requires a membrane receptor for 1,25(OH)_2_D_3_ and the subsequent activation of secondary messengers such as cAMP or calcium (recently reviewed in: [66]). Protein disulfide isomerase (PDIA3), also known as pER57 or 1,25D_3_-MARRS (membrane-associated, rapid response steroid-binding) is the most studied candidate for membrane vitamin D receptor, although a detailed mechanism of interaction between 1,25(OH)_2_D_3_ and PDIA3 is not fully understood [66,67,68]. Recent studies also provided evidence that mitochondria could be a direct target of 1,25(OH)_2_D_3_ [69,70]. This observation may support previous studies showing the protection of mitochondria by 1,25(OH)_2_D_3_ by the modulation of the levels of oxidative stress (e.g., mitochondrial membrane potential) and the expression of genes thata re involved in response to reactive oxygen species [71,72,73]. Interestingly, it seems that mitochondrial localization of VDR protect mitochondrial from oxidative and nitrosative stress [72]. On the other hand, in vitro results suggested that the mitoprotective effects may depend on the concentration of active analogs of vitamin D, the time of incubation, or are cell-type-specific [71,74]. The mitoprotective effect of vitamin D and its analogs was also observed in skin cells that were subjected to ultraviolet light [70,73,75,76,77,78,79]. Interestingly, the mitochondria could also be the targets for anticancer and/or inflammatory activities of vitamin D and its analogs as well as derivatives of lumisterol or other related steroidal analogs [76]. The malfunction of mitochondria and the excessive production of ROS contributes to inflammation that is characteristic for psoriasis [80]. Furthermore, the impairment of the mitochondrial-induced apoptotic pathway may also result in hyperproliferation of keratinocytes [81]. Thus, it seems that the direct, nongenomic impact of 1,25(OH)_2_D_3_ on cellular processes including mitochondrial function may contribute to its anti- psoriatic activities of this powerful hormone [66].

Our previous study showed that vitamin D in addition to activation by 25 and 1-hydroxylations, can also be activated by the rate limiting enzyme of steroidogenesis, CYP11A1 [82,83,84,85], with the generation of 20(OH)D_3_, as the first and main product of this pathway. 20(OHD)_3_ can be further hydroxylated by other enzymes together with downstream metabolites [83,86,87,88]. This pathway functions in vivo in humans and animals and can act on a local and systemic level [82,83,86,89]. 20(OH)D_3_ and its metabolites without OH at C1α can act as a biased agonist on VDR as indicated by the lack of calcemic effects and the poor activation of CYP24A1 [86,90,91], and by studies on ligand-induced VDR translocation to nucleus and molecular modeling [86,92,93,94] and crystallography using the ligand binding domain of the VDR [95,96]. In addition, novel pathways of lumisterol [97,98] and tachysterol [99] activation have been discovered.

Recently, we also showed that 20(OH)D_3_, 20,23(OH)_2_D_3_ and their metabolites, that were generated by alternative pathway, can act as inverse agonists on retinoic acid-related orphan receptors, RORα and RORγ [100,101,102], which belong to the ROR subfamily of nuclear receptors, that play a crucial role in the variety of physiological processes, including immune functions [103,104,105]. In addition, lumisterol hydroxyderivatives act as inverse agonists on RORs [97]. Importantly, aryl hydrocarbon (AhR) was identified as an alternative receptor for vitamin D hydroxyderivatives [102,106,107]. In addition, hydroxyderivatives of vitamin D and lumisterol compounds act as ligands on liver X receptors (LXR)α and β [102,108]. Thus, there is more than one bioactive form of vitamin D and several additional to the VDR nuclear receptors that are activated by these compounds [36]. In addition, it has been documented that lumisterol, a photoderivative of vitamin D_3_, can be activated to the biologically active hydroxyderivatives that would act on LXRs and RORs to exert their phenotypic effect [97,105,108,109]. Vitamin D and lumisterol hydroxyderivatives can also interact with SARS-CoV-2 replication machinery enzymes [110] and angiotensin-converting enzyme 2 (ACE2) and TMPRSS2 [111] and their protective role in COVID-19 has been discussed [112].

## 4. Vitamin D and Epidermal Keratinocytes

Keratinocytes express VDR and RORs and can produce and metabolize 1,25(OH)_2_D_3_ [101,113]. Skin cells also express CYP enzymes that metabolize vitamin D_3_ pro-hormone to its biologically active form [45,75,114,115,116,117] (Figure 2). VDR expression and VDR-mediated signaling pathways in the skin are essential in maintaining the skin homeostasis [118,119,120,121]. The expression of markers of differentiation (involucrin, profilaggrin, and loricrin) are decreased the interfollicular epidermis and the hair follicle in VDR knockout mice model [121].

Hosomi et al. [122] firstly reported the stimulatory effects of 1,25(OH)_2_D_3_ on keratinocyte differentiation by inhibiting DNA synthesis, thus decreasing the number of cells, increasing the density and the size, and differentiation into squamous and enucleated cells, and the stimulation of the formation of a cornified envelope. These findings were confirmed by others [123,124,125]. The effects of VDR-mediated pathways depend on coactivators and corepressors [126,127]. Vitamin D, acting through VDR, and DRIP205, SRC2, and SRC3 coactivators can stimulate the keratinocytes differentiation markers: expression of involucrin, loricrin, filaggrin, keratins, and transglutaminase activity [128]. Bikle et al. [127] reported that the main coactivators are the vitamin D interacting protein (DRIP) complex that is involved mainly in the proliferation of keratinocyte and the steroid receptor coactivator (SRC) complexes that are involved in keratinocytes differentiation. They found also that DRIP205 plays a role in the regulation of β-catenin pathways, including cyclin D1 and Gli1 expression, and SRC3 can regulate lipid synthesis, permeability barrier formation that is related to differentiation. The effects of vitamin D on keratinocytes could be dependent on other culture conditions. Gniadecki et al. reported that in cultures of 1,25(OH)_2_D_3_ at the concentration from 10^−11^ to 10^−6^ M and 0.15 mM calcium in the absence or with low levels (0.1 ng/ml) of epidermal growth factor, the keratinocytes cell cycle was blocked in the late G1 phase, while the culture of keratinocytes with 1,25(OH)_2_D_3_ at the concentration of 10^−11^ to 10^−9^ M and high extracellular calcium concentration (1.8 mM) stimulated cell growth was observed (increasing the proportion of cells entering the S phase) [129].

20(OH)D_3_ inhibits proliferation, causes G1/0 and G2/M arrest, and stimulates differentiation of keratinocytes [125] and inhibits NFκβ [130]. Correspondingly, 20*S*(OH)D_3_, acts through VDR (stimulates its expression), inhibits growth, inhibits DNA synthesis. It also stimulates the expression of involucrin—differentiation marker for keratinocytes [125,131]. Other CYP11A1 vitamin D derivatives show similar anti-proliferative and pro-differentiation properties [88,93,102,132,133,134,135]. Similarly, vitamin D derivatives (resulted from CYP11A1 activity) protect from UV-induced DNA damages by the activation of Nrf2 and p53 defense mechanisms [73,77,134,136] and have shown anti-tumor activity against epidermal cancers [75,137]. Vitamin D derivatives have also shown antiproliferative activity on melanocytes [138,139] and fibroblasts with anti-fibrogenic actions [140,141,142,143], being dependent on RORγ [144]. In addition, vitamin D derivatives increased the expression of hypothalamic-pituitary-adrenal axis neuropeptides: CRF, urocortins and POMC, and their receptors, CRFR1, CRFR2, MC1R, MC2R, MC3R, and MC4R human epidermal keratinocytes [145]. Thus, vitamin D derivatives, VDR, and its coactivators are important for the epidermis differentiation and maintenance. Finally, 20(OH)D_3_ has recently been shown to have therapeutic effects in in vivo models of rheumatoid arthritis (RA) [146,147].

## 5. The Active Forms of Vitamin D Act as Powerful Immunomodulators

The active forms of vitamin D, acting through VDR, modulates the maturation, activity, and functions of monocytes, macrophages, T- and B-cells, and dendritic cells (DCs) [148]. In general, vitamin D promotes the innate immune responses by enhancing phagocytic functions of immune cells, while inhibiting the adaptive immune system. The activation of monocytes and macrophages by biologically active vitamin D derivatives resulted in the increased production of cathelicidin antimicrobial peptide (CAMP) with its processing to LL 37. In dendritic cells presenting antigens, vitamin D decreases the maturation, expression of MHC Class II molecules, co-stimulatory molecules (CD40, CD80, and CD86) and IL-12, and increases IL-10 production (reviewed in [149]). Vitamin D decreases the development of human natural killer (NK) cells and inhibits the cytotoxicity and cytokine production by developed NK cells, while in hematopoietic stem cells stimulated the expression of monocytes markers (C/EBPα and CD14) [150]. Vitamin D acts as an immunosuppressive molecule decreasing the proliferation and functions of T lymphocytes that is mediated by inhibiting IL-2 and IFNγ production [151,152]. Vitamin D inhibits polarization towards Th1 cells [153,154] and pro-inflammatory Th1-related cytokines by T lymphocytes, as IL-2, IFNγ, and TNFα [153,155,156,157,158,159]. The development of Th2 cells of Th lymphocytes, strong polarization toward a Th2 profile is also enhanced by vitamin D in the IL-4-depended pathways. The neutralization of IL-4 abolishes vitamin D_3_-induced polarization of Th2 independently of IFN-γ [160]. In addition, vitamin D_3_ enhances the expression of Th2-specific transcription factors, GATA-3 and c-maf, in developing Th cells [160]. Vitamin D promotes the immunosuppressive response by enhancing the activity of CD4^+^ CD25^+^, mostly expressing FoxP3, without the changing the number of CD4CD25Foxp3 cells [161]. Similarly, in a colitis mouse model, vitamin D suppressed the immune response (downregulates IFNƴ, TNFα, MPO activity, and IL-1β), downregulated the expression of T-bet (Th1 transcription factor), and up-regulated the expression of GATA3 and IL-4 [162].

Not only does the classical vitamin D pathway show anti-inflammatory properties, but also vitamin D derivatives suppress inflammatory responses [101,133,134,146,147]. Chaiprasongsuk and co-workers reported that CYP11A1-derived vitamin D_3_-hydroxyderivatives, as 20(OH)D_3_, 1, 20(OH)_2_D_3_, 20,23(OH)_2_D_3_, and 1, 20,23(OH)_3_D_3_ modulated the expression of inflammation-related genes, downregulating 11 out of 16 genes, including *NFkB p65* (*Rel A*), and up-regulating 5 out of 16 genes, including *NFkB p50,* and *IkB-α* [133,134]. The key factor that is involved in the anti-inflammatory action of vitamin D derivatives is NFkB. Janjetovic at al revealed that in keratinocytes 20(OH)D_3_ decreases NF-κB activity by increasing IκBα levels [130], similarly to 20,23(OH)_2_D_3_ [132]. 20(OH)D_3_ also inhibits the production of pro-inflammatory cytokines TNFα and IL-6 [133]. Furthermore, in DBA/1 Lac J CIA model of rheumatoid arthritis, the inhibition of Th1 and Th17-related cytokines production by CYP11A1-derived secosteroids was found [133,147].

The inverse correlation between vitamin D and the pathogenesis of autoimmune diseases, including psoriasis has been published [163]. Vitamin D regulates the function of the innate and adaptive immune response, thus representing a potential protectant and therapeutic for psoriasis. The effects of vitamin D on the immune system in psoriasis are complex. For example, Vitamin D promotes differentiation of naïve T-cell differentiation into T regulatory cells, thus enhancing the production of anti-inflammatory cytokines (TGF-β, IL-4, and IL-10), and suppressing the production of pro-inflammatory cytokines (TNFα, INFƴ, IL-2, IL-17A, and IL-21) (reviewed in [148]). Calcipotriol decreased the frequency of CD8^+^ IL-17^+^ T-cells in psoriatic lesions [164]. The activity of DCs, DCs-mediated induction of T-cell proliferation, Th1 cytokine IFN*γ* production is suppressed by vitamin D [165]. Vitamin D inhibited the IL_17-induced expression of IL-1Ra, IL-36α, IL-36β, and IL-36*γ,* and the TNF-α-induced expression of IL-1Ra, IL-36Ra, IL-36α, IL-36*γ,* and beta-defensin 2 (HBD2) in human keratinocytes [166]. Besides, in psoriasis calcipotriol decreased the Th17 cytokine-mediated pro-inflammatory S100 psoriasin (S100A7) and koebnerisin (S100A15) [167] and HBD2 and HBD3, IL-17A, IL-17F, and IL-8 production. The inhibition of IL-17A induced-HBD2 expression was mediated by increasing IkappaB-α protein and the inhibition of NF-κB signaling, while VDR and MEK/ERK signaling pathways were activated and involved in the induction of cathelicidin [168]. Interestingly, vitamin D also stimulates the expression of IL-33 and its receptor ST2 [169] and IL-33 was show to alleviate Th17-mediated psoriatic inflammation [170]. Thus, the anti-inflammatory activity of vitamin D is an important factor that is useful in the pathogenesis and management of psoriasis.

## 6. Vitamin D and Psoriasis

### 6.1. Vitamin D Serum Level in Psoriatic Patients

The results of several epidemiologic studies have identified the correlation between the vitamin D serum level and the likelihood of some diseases development or progression in patients with low or deficient levels of vitamin D [112,171,172,173,174]. Similar studies have been published for psoriasis [175,176,177,178]. Morimoto and co-workers showed that the 1,25-dihydroxyvitamin D (1,25(OH)2D, but not 25-dihydroxyvitamin D (25(OH)D), was inversely correlated to area-severity index and the severity index in psoriasis patients. In these patients, the level of vitamin D was in the normal range [179]. Tajjour et al. reported decreased 25(OH)D levels in psoriasis patients and a negative correlation with the severity of the disease [180]. Bergler-Czop and Brzezinska-Wcislo reported a lower level of 25(OH)D in the psoriasis group than in the control group, with a deficient level in psoriasis and insufficient level in the control group [181]. Furthermore, the level of 25(OH)D was negatively correlated to PASI score and the duration of psoriasis [181,182]. The lower level of 25(OH)D was also found in case-control studies [183,184,185]. In addition, Filoni et al. also found a reduced level than in the control group and the correlation between vitamin D levels and psoriasis duration [175]. Similarly, Grassi and co-workers in a cross-sectional study observed a lower free and total vitamin D serum level in chronic plaque psoriasis patients than in the controls [176]. The relationship between vitamin D and psoriasis was also confirmed in a meta-analysis [177]. However, still the nature of this correlation is unclear and further studies are needed to elucidate its role in the pathogenesis of psoriasis, and still it is not known if low vitamin D level is the causative factor for psoriasis or the effect of the disease. On the contrary, some reports showed no differences in the 25(OH)D serum level between psoriasis and the control group [186]. Mattozzi et al. found the positive correlation between the vitamin D serum level and Tregs, and suggested that a decreased level of vitamin D may promote the activity of Th1, Th17, and Th22 [14]. They also found a negative correlation between PASI score and vitamin D level, but only Tregs were significantly related with vitamin D in multiple regression analysis [14]. The decreased level of vitamin D is negatively correlated to inflammatory activation marker—C-reactive protein [183]. A meta-analysis of 10 published reports and 571 psoriatic patients 496 controls confirmed the reduced 25(OH)D level in the disease group and negative correlation between circulating 25(OH)D levels and PASI score [187]. Vitamin D deficiency has been suggested as one of the environmental factors that is involved in psoriasis as immune-mediated disorder. Some studies have confirmed that vitamin D deficiency can be found in psoriatic patients being associated with the severity of a disease.

### 6.2. On the Link between UVB Phototherapy, Serum 25(OH)D Levels and Psoriasis Natural History

The synthesis of vitamin D in the skin starts with the conversion of 7-dehydrocholesterol to previtamin D_3_ after the absorption of UVB. The phototherapies of psoriasis are based on UVB (290–320 nm), narrowband UVB (NB-UVB) (311 nm), excimer laser (308 nm), UVA1 (340–400 nm), psoralen and UVA (PUVA, 320–400 nm), and others (reviewed in [188]. The studies revealed that NB-UVB therapy has an impact on the systemic serum level of vitamin D in psoriasis patients. In psoriasis, patients that were treated with NB-UVB, the increase of the vitamin D serum level from insufficient to normal range was found, but no relationship with PASI score and/or SCORAD improvement [189,190,191]. Similarly, Ryan et al. noted that the serum 25(OH)D level increase from median of 23 ng/mL at to 51 ng/mL at the end of NB-UVB, with no correlation with treatment response [192]. Ala-Houhala et al. published the data that were related to NB-UVB and oral supplementation of cholecalciferol, 20 μg daily. The psoriasis patients and healthy controls increased 25(OH)D level found similar results [193]. NB-UVB exposure did not change the expression of CYP27A1 and CYP27B1 in psoriasis patients, while in the healthy controls its expression decreased. In healthy controls the expression cathelidicin decreased, HBD2 increased slightly, while in psoriasis patients cathelidicin expression did not changed, while HBD2 expression decreased [193]. The increase of 25(OH)D after UVB-based psoriasis therapy is also observed after UVA/NB-UVB treatment [194,195]. A similar trend was found also for 25(OH)D and for broadband UVB (BB-UVB) therapy, with BB-UVB showing strongest effect [196]. It should be noted that UVA1 therapy decreased the 25(OH)D serum level from 21.9 to 19.0 ng/mL [194]. The increase of the vitamin D serum level after NB-UVB treatment is accompanied by changes in athe ntimicrobial peptide and cytokine expression (increasing expression of cathelicidin and decreasing levels of human beta-defensin 2) [190], but these changes are related to the season of the irradiation [197]. Thus, the balance between vitamin D and the expression of antimicrobial peptides could be involved in the therapeutic effects of NB-UVB. On the contrary, Vandicas et al. [198] noticed a higher level of 25(OH)D and vitamin D binding protein (VDBP), acting as transporter and reservoir for vitamin D and its metabolites [199,200] in psoriasis patients. In addition, as in other studies, 25(OH)D increased after UVB treatment, while VDBPs were not changed and did not correlate with 25(OH)D levels. Thus, the authors suggested that VDBPs could be a marker of systemic inflammation. However, immunosuppressive effects of the UVB can also be secondary to the activation of the central [201,202,203,204] or local neuroendocrine networks [26,28,35,205,206].

### 6.3. Local Vitamin D Endocrine System in Psoriasis

Vitamin D acts mainly through VDR [101,119,207], and the disturbances of its expression have been observed in several diseases (for example: [208,209,210,211]). The expression of vitamin D receptors have been found in psoriatic cells (Figure 2) [212]. Since vitamin D can regulate the proliferation and growth of keratinocytes, the impairment of VDR expression in epidermal skin cells could be involved in the pathogenesis of psoriasis [119,213,214,215]. The first reports did not show the changes in vitamin D system and reported comparable VDR levels and receptor binding to DNA in psoriatic and normal skin [216]. Milde et al. reported no differences in VDR expression between normal and non-lesional skin, but found stronger VDR expression in psoriasis when compared to non-lesional skin [217]. A very recent study showed that VDR is expressed in psoriatic skin and shows mainly strong expression, especially in the basal layer (Chandra, Roesyanto-Mahadi et al., 2020). On the other hand, Kim et al. [218] found reduced VDR expression in psoriasis and perilesional skin than in normal skin. They also reported the negative correlation between Toll-like receptor 2 (TLR2) and VDR expression in psoriasis, a negative correlation between TLR2 and VDR expression in the psoriasis skin of vitamin D-deficient groups, but a positive correlation in psoriasis skin of a vitamin D-sufficient group [218]. The authors concluded that psoriasis patients, according to vitamin D serum level, could be treated differently with therapies that modulated the TLR-VDR pathways. Similarly, Visconti et al. also observed the reduced VDR expression in psoriasis, with the preserved expression in deeper layers of the epidermis [219]. Contradictory data on VDR expression could result from methodological issues and using of different anti-VDR antibodies. The pathogenic effects of the potential disturbed expression of VDR in keratinocytes could result in keratinocytes adhesion. Visconti et al. found significantly reduced expression of occludin and claudin 1 (proteins forming tight junctions) than in normal skin. Furthermore, the percentage of claudin-1- and zonulin-1-positive cells (proteins forming tight junctions) correlated to the percentage of VDR-positive cells [219]. The authors suggested that VDR expression in the skin is essential to the preservation of skin integrity and homeostasis by forming tight junctions [219].

Not only the changes of the expression of VDR are probably linked to psoriasis but also VDR polymorphism. A meta-analysis of 11 studies revealed that FokI and ApaI *VDR* polymorphisms are not linked to psoriasis risk, but the BsmI B variant shows borderline association [220]. For Caucasians, the TaqI t variants were allied with reduced psoriasis risk [220]. Similarly, a meta-analysis of 16 studies showed that *VDR* TaqI TT variant in Caucasians but not in Asians is related to a higher risk of psoriasis, but *VDR* ApaI, BsmI,t or FokI polymorphisms have no association with the disease [221]. On the other hand, A-1012G *VDR* promoter polymorphism and Fok1 were identified as related to the susceptibility to non-familial psoriasis [65]. A recent study showed that the TaaI/Cdx-2 GG genotype, related to regulation of IL-17 and IL-23 expression, are more frequent in psoriasis patients [222].

The novel vitamin D derivatives, 20(OH)D_3_ and 20,23(OH)_2_D_3_, can act as the inverse agonists on RORα and RORγ, which are expressed in human skin, keratinocytes, fibroblasts, melanocytes, and others [101,135]. Previously, we reported the elevated expression of RORγ in lymphocytes in psoriasis skin [212]. RORγ is a key factor that is involved in the differentiation of lymphocytes into IL-17-producing Th17 cells [223]. In addition, RORα and RORγ can regulate IL-17 expression [103,224]. 1,25(OH)_2_D_3_, 20(OH)D_3_ and 20,23(OH)_2_D_3_ inhibited the RORα and RORγ-mediated activation of the *IL17* promoter in a dose-dependent manner and production of IL-17 protein [101]. Since IL-17 and TH17 are the crucial factors in psoriasis pathogenesis, they represent a molecular target [225,226]. A-9758, selective ROR*γ*t inverse agonist, efficiently inhibited IL-17A release in in vitro and in vivo models [226]. A selective RORγt inhibitor Cpd A inhibited the transcriptional activity of human RORγt, differentiation of Th17 cells and the production of pro-inflammatory cytokines by T-cells, IL17F, IL22, IL26, IL23R, and CCR6 [227]. Another molecule, SR1001, synthetic RORα/γ inverse agonist in mouse models of atopic dermatitis and acute irritant dermatitis showed the anti-inflammatory effects (decreased the expression of IL-13, IL-5, IL-17A) and restored keratinocytes differentiation [228]. RORγt is also an important transcription factor regulating IL-22 in Th22 cells, which are related to inflammation and linked to the pathophysiology of psoriasis. 1,25(OH)_2_D_3_ can regulate the expression of IL-22, since vitamin D response elements have been identified in the *IL22* promoter [229].

## 7. Vitamin D Paradigm in the Psoriasis Treatment

### 7.1. Introduction to the Problem

Since vitamin D regulates the immune system functions and the proliferation and differentiation of the keratinocytes, as well as other cell types, it has been effectively incorporated as an adjuvant treatment for psoriasis [230,231,232], although the precise mechanism of therapeutic action of vitamin D is not fully known. The first reports showing the therapeutic potential of vitamin D were published almost 90 years ago [233,234], but the use of high doses of vitamin D was limited due to potential toxic effects. The next important step was done by Morimoto S, who published several papers confirming the therapeutic effects of orally or topically administrated 1α(OH)D_3_ or 1,25(OH)_2_D_3_ [235,236,237,238,239]. The significant response of the lesional psoriatic skin to the treatment was observed in up to 85% of psoriasis cases [235,236,237,238,239]. The authors also showed the inhibitory effects of analogues of 1,25(OH)_2_D_3_ with no effects of 1,25(OH)_2_D_3_ on cultured psoriatic fibroblasts [240]. MacLaughlin et al. also observed the partial response of psoriatic fibroblasts to 1,25(OH)2D3, but only at higher tested doses (10 and 100 µM) [241]. The studies also confirmed the safety of long-term psoriasis treatment that was based on vitamin D [242,243,244]. The further studies allowed to develop the effective anti-psoriatic treatment that was based on vitamin D derivatives, including 1,25(OH)_2_D_3_, calcipotriol, maxacalcitol, tacalcitol, hexafluoro-1,25(OH)_2_D, calcipotriene, and others (reviewed in [230,245].

It is accepted that the therapeutic effects of vitamin D and its derivatives require *VDR* expression since above mentioned processes are mediated by this receptor and that *VDR* polymorphisms could modulate the responsiveness to the treatment. Recently, it has been reported that psoriasis patients with a PASI score that is lower than 3 and the rs7975232 CC genotypes were much more susceptible to calcipotriol treatment [246]. Lesiak et al. did not find the correlation of the rs7975232 variant and the treatment, but they observed the relationship between different variants of TaaI/Cdx-2 and the effects of UVB-based treatment, as assessed by the analysis of cytokines expression (IL-17, IL_23, TNF alpha) [222]. Taq1 *VDR* polymorphism (rs731236) was found to be a predictor of the duration of remission with C allele homozygotes that were related to the decreased VDR activity, showing a shorter remission duration than heterozygotes and T allele homozygotes [247]. A-1012G promoter polymorphism, and F and T alleles of Fok1 and Taq1 polymorphisms of *VDR* have been also identified as positively with calcipotriol response: AA and TT genotypes and AAFF, AATT, and FFTT genotype combinations [65]. On the contrary, BsmI and ApaI *VDR* polymorphisms are correlated to responsiveness to calcipotriol in Korean psoriasis patients [248].

It is well accepted that the topical application of vitamin D and/or its derivatives can improve psoriasis. However, psoriasis is systemic disease, thus systemic treatment should also be applied.

### 7.2. Oral Treatment with Vitamin D and Its Derivatives

The first vitamin D derivative that was used for psoriasis treatment by oral administration was 1α(OH)D (1.0 µg/day for 6 months) [235,237,238]. Its therapeutic action could be result from the inducing the changes in keratins expression. Holland and co-workers observed lowering the expression of keratin 16 and keratin 2 overexpression after 1α(OH)D treatment [249]. Thus, 1α(OH)D is able to inhibit the keratinocytes proliferation and promote its differentiation and shows anti-psoriatic properties. The oral administration of 2 µg/day of 1,25(OH)_2_D_3_ in the pilot study of Huckins et al. also resulted in a significant improvement of psoriatic arthritis [250]. Supplementation with vitamin D 5000 IU/day for three months significantly increased the vitamin D serum level, and the expression of anti-inflammatory cytokines (IL-10, IL-5) and decreased the PASI score and homocysteine plasma level, the expression of pro-inflammatory cytokines (IFN-7, TNF-α, IL-1β, IL-6, IL-8, and IL-17) and high-sensitivity C-reactive protein [251]. Similarly, a double-blind, randomized, placebo-controlled study with oral administration of vitamin D2 60,000 IU once every 2 weeks for 6 months resulted in an improved PASI score and an increase of 25(OH)D serum level with no signs of adverse effects. In addition, the serum level of 25(OH)D was negatively correlated to the PASI score [231].

The earlier studies used relatively low doses of vitamin D or its derivatives. But Finamor et al. in their study used 35,000 IU of vitamin D_3_/day for six month and observed a significant 25(OH)D3 increase and improvement of PASI score, with no signs of toxicity: no change of serum urea, creatinine, and calcium and a change of urinary calcium excretion within the normal range [252]. McCullough et al. studied the effects of long-term high doses of vitamin D_3_ administration (up to 50,000 IU) in patients that were affected by psoriasis with a significant improvement observed with no toxicity and adverse effects related to vitamin D treatment [253]. These results indicated that high doses of vitamin D could be efficient and safe in psoriasis treatment.

Some results are inconclusive or present contradictory data. Studies of Ingram et al. concerned the effects of vitamin D_3_ at the doses of 100,000 IU/month for 12 months (200,000 IU at baseline) (Australian New Zealand Clinical Trials Registry #12611000648921 [254]). They observed no changes of the PASI score, but the level of 25(OH)D increased. Additionally, an inverse correlation between a slight decrease of PASI score and 25(OH)D (up to 125 nmol/L) was also found. Similarly, Jarrett et al. [255] did not recommend the administration of vitamin D_3_ (100,000 IU per month) to treat the psoriasis, since no significant differences were observed between the supplemented and the placebo groups. In study of Prystowsky et al., no additive effect of orally administrated calcitriol (0.5–2.0 µg/day) on UVB phototherapy of psoriasis was found [256].

In summary, the meta-analysis of randomized controlled trials of oral vitamin D supplementation in psoriasis patients confirmed the effectiveness of the improvement the PASI score, however, Hartung-Knapp adjustment these effects were not significant [257]. Thus, more randomized controlled trials, especially with the use of vitamin D derivatives are needed. Furthermore, more effective treatment is related with topical administration. A study by Gumowski-Sunek and co-workers found changes in calcium metabolism after the oral administration of calcitriol (1.5 µg/day), while the topical administration of calcipotriol with an equivalent dose 150 µg/day, with 1% absorption) did not change calcium metabolism [258].

### 7.3. Topical Treatment with Vitamin D and Its Derivatives

The topical treatment is a first-line therapy in patients with mild or moderate psoriasis [259]. Early reports of Morimoto et al. showed that the topical administration of 0.1 and 0.5 µg/day of 1,25(OH)_2_D_3_ resulted in a significant improvement of psoriatic lesions with no toxicity symptoms [239]. For the topical psoriasis treatment, the calcipotriol has been introduced as an efficient and safe adjuvant already in late 1980s [260,261]. The treatment with calcipotriol resulted in the reduced expression of IL-6, with no changes in TNF alpha expression [262]. The efficacy of UVB treatment of psoriasis with ointment containing of calcipotriol (50 µg/g ointment twice a day) was greater than UVB alone [263]. Calcipotriol-containing ointment (50 µg/g twice a day) treatment also improved the psoriasis therapy with fumaric acid ester in multicenter, randomized, double-blind, vehicle-controlled study [264], psoriasis treatment with cyclosporine [265], and psoriasis treatment with acitretin [266]. Similarly, calcipotriene (0.005% and betamethasone dipropionate 0.064%) ointment improved the presentation of psoriasis [267]. Pinter et al. data showed the high effectiveness of calcipotriene and betamethasone PAD^TM^ Technology cream with no adverse drug reaction in two Pahase 3, multicenter, randomized, investigator-blind studies [268]. Different clinical studies report that the combination of calcipotriene and betamethasone dipropionate in foam solution to be the most effective treatment for mild psoriasis [269].

Equally, a multicenter study concerning the use of tacalcitol (4 µg/g) ointment once daily for 18 months showed its high effectiveness, safety, and good tolerance during the long-term topical treatment, with no changes in calcitriol, calcium, and parathyroid hormone serum level [270]. The multicenter prospective study with tacalcitol (20 µg/g) ointment applied once daily revealed the decrease of PASI score, with local adverse effects, a decrease of parathyroid hormone and 1,25(OH)_2_D_3_, but a maintained serum calcium homeostasis was observed [243]. Tacalcitol also increased the effectiveness of NB-UVB treatment [271]. Thus, tacalcitol is effective and safe during long-term treatment.

Maxacalcitol, another vitamin D derivative, is very effective in inducing of differentiation and inhibiting of keratinocyte proliferation, without inducing hypercalcemia [272]. These effects were stronger than that of either calcipotriol or tacalcitol in in vitro models. A placebo-controlled, double-blind study showed that maxacalcitol ointment (6, 12.5, 25, and 50 mg/g) was very effective in the improvement of psoriasis presentation, with 25 mg/g ointment being more efficient than calcipotriol [273], especially in combined treatment that included maxacalcitol and NB-UB [274]. However, use of maxalcitol is related to a higher risk of development of hypercalcemia, and psoriasis treatment with calcipotriol is safer in comparison to maxalcitol [275].

The topical application vitamin D and its derivatives constitutes an important element of psoriasis management and can also be used in combination with other modalities. The topical treatment offers a safer therapy option, since the active compounds can directly target the lesional areas, without excessive systemic entry.

## 8. Conclusions and Future Directions

Vitamin D and its derivatives have resulted in impressive clinical responses in psoriatic patients. They represent effective and safe adjuvant treatments for psoriasis, even when high doses of vitamin D are administered. The phototherapy of psoriasis, especially UVB-based, changes the serum level of 25(OH)D. However, such a correlation of 25(OH)D levels and psoriasis improvement requires future clinical trials since contradictory data have been published in this area. Vitamin D derivatives can improve the efficacy of psoriasis UVB phototherapy without inducing adverse side effects. Excellent candidates for anti-psoriatic treatment are new, none, or low calcemic CYP11A1-derived hydroxyderivatives of vitamin D_3_ or lumisterol compounds. These would act as agonists on VDR, LXR, and AhR receptors and as inverse agonists on RORs. In conclusion, targeting of local vitamin D signaling systems represents a promising future in therapy or prevention of psoriasis.

## Figures and Tables

**Figure 1 ijms-23-08575-f001:**
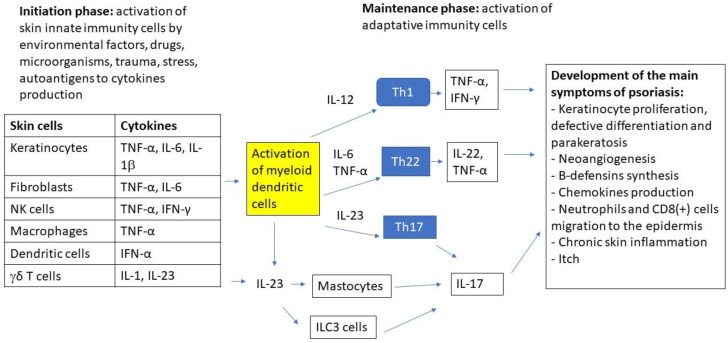
The major effector cells and signaling pathways in the immunopathogenesis of psoriasis.

**Figure 2 ijms-23-08575-f002:**
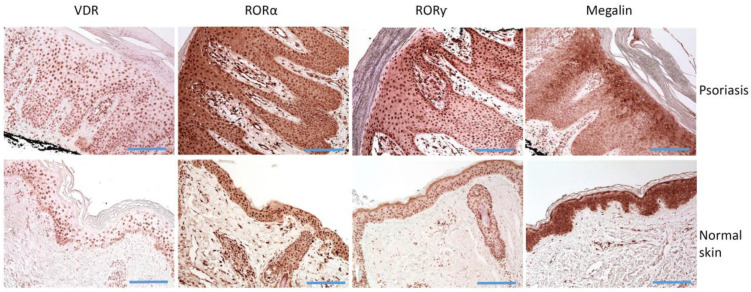
Immunostaining of VDR, RORα and RORγ, and LRP2/megalin in psoriasis and normal skin. Histology of lesional skin shows skin with regular psoriasiform epidermal hyperplasia, prominent dilated vessels with an edematous dermal papillae, mounds of parakeratosis containing neutrophils, and a superficial perivascular lymphocytic infiltrate. Scale bars = 100 μm.

## Data Availability

Not applicable.

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
