# Peer review of "Vitamin D Signaling in Psoriasis: Pathogenesis and Therapy"

_ijms, 2022, doi:10.3390/ijms23158575_

Round 1

Reviewer 1 Report

The manuscript entitled “Vitamin D signaling in psoriasis: pathogenesis and therapy” is a review paper describing the effect of vitamin D administration in psoriasis treatment. It is a comprehensive review with subtitles explaining the psoriasis manifestation, involved signaling pathways in disease pathogenesis, and role of vitamin D and its derivative as a treatment approach, and the vitamin D-associated signaling pathways. I have some comments/amendments that the authors should address before the publication. Several typos and English mistakes in the text and Figure 1 should be corrected (some of them are mentioned below). In Figure 2, the authors did not mention the difference in histopathology of psoriasis compared to normal skin.

Here are examples of multiple typo mistakes:

Figure 1 “maintenance

page 2, line 74 “…the differentiation Th0 cells…”

page 4, line 144 “… ant inflammatory activities of…”

page 4, line 167 “…as an alternative receptor fro…”

page 5, line 198 ”… They found also that DRIP205 also plays…”

page 5, lines 217-218 “….In addition, vitamin D and its 217 derivatives increased the expression of expression of hypothalamic-pituitary-adrenal axis…”

page 6, line 239 “…Vitamin D inhibits poliarization towards Th1 cells…”

page 6, line 246 “…in majority expressing FoxP3, without the changing the…”

page 6, line 255 “…showing downregulation 11 out of 16…”

page 6, line 256 “…and up-regulation 5 out of 16…”

page 7, line 309 “…but only Tregs was significant related with vitamin D in multiple regression analysis…”

page 7, line 314 “…Vitamin D deficiency has been suggested as one of environmental factor that could be…”

page 9, line 397 “Il17 promoter” and line 409 “il22 promoter” Human gene symbols should be in capital letters.

page 11 Lines 481-483 “…In study of Prystowsky et al. no additive effect of orally administrated calcitriol (0.5-2.0μg/day) on UVB phototherapy od psoriasis was found…”

page 12, lines 540-541 “…Thus, targeting vitamin D signaling would represent promising futhure in therapy or prevention of psoriais…”

Author Response

Reviewer 1

The manuscript entitled “Vitamin D signaling in psoriasis: pathogenesis and therapy” is a review paper describing the effect of vitamin D administration in psoriasis treatment. It is a comprehensive review with subtitles explaining the psoriasis manifestation, involved signaling pathways in disease pathogenesis, and role of vitamin D and its derivative as a treatment approach, and the vitamin D-associated signaling pathways. I have some comments/amendments that the authors should address before the publication. Several typos and English mistakes in the text and Figure 1 should be corrected (some of them are mentioned below).

Reply:

We thank the reviewer for his/her attention and positive comments. We corrected the manuscript with Track changes option.

In Figure 2, the authors did not mention the difference in histopathology of psoriasis compared to normal skin.

Reply:

Thank you for this comment. We corrected the figure legend:

Figure 2. Immunostaining of VDR, RORα and RORγ and LRP2/megalin in psoriasis in psoriasis and normal skin. Histology of lesional skin shows skin with regular psoriasiform epidermal hyperplasia, prominent dilated vessels with an edematous dermal papillae, mounds of parakeratosis contain-ing neutrophils, and a superficial perivascular lymphocytic infiltrate. Scale bars = 100 μm.

Here are examples of multiple typo mistakes:

Figure 1 “maintenance

page 2, line 74 “…the differentiation Th0 cells…”

page 4, line 144 “… ant inflammatory activities of…”

page 4, line 167 “…as an alternative receptor fro…”

page 5, line 198 ”… They found also that DRIP205 also plays…”

page 5, lines 217-218 “….In addition, vitamin D and its 217 derivatives increased the expression of expression of hypothalamic-pituitary-adrenal axis…”

page 6, line 239 “…Vitamin D inhibits poliarization towards Th1 cells…”

page 6, line 246 “…in majority expressing FoxP3, without the changing the…”

page 6, line 255 “…showing downregulation 11 out of 16…”

page 6, line 256 “…and up-regulation 5 out of 16…”

page 7, line 309 “…but only Tregs was significant related with vitamin D in multiple regression analysis…”

page 7, line 314 “…Vitamin D deficiency has been suggested as one of environmental factor that could be…”

page 9, line 397 “Il17 promoter” and line 409 “il22 promoter” Human gene symbols should be in capital letters.

page 11 Lines 481-483 “…In study of Prystowsky et al. no additive effect of orally administrated calcitriol (0.5-2.0μg/day) on UVB phototherapy od psoriasis was found…”

page 12, lines 540-541 “…Thus, targeting vitamin D signaling would represent promising futhure in therapy or prevention of psoriais…”

Reply:

Thank you for this comment. We corrected the manuscript with Track changes option.

Reviewer 2 Report

I read with great interest this review article toward vitamin D and psoriasis.

It is remarkable the literature evaluation  and it is solidly described.

I have only some minor revision to add:

Please change the reference number 1 that is a link to a non governative web site and change it with GBD-2019 psoriasis related data published on Frontiers in Med.

Please mention carefully the cardiovascular implications of inflammatio based low levels of vitamin D.

Please describe the cardiovascular comorbidities in psoriasis with a particular attention the endotypes [10.3390/jcm9010186, 10.1016/j.jid.2019.07.727]

Author Response

Reviewer 2

I read with great interest this review article toward vitamin D and psoriasis.

It is remarkable the literature evaluation  and it is solidly described.

Reply:

We thank the reviewer for his/her attention and positive comments.

I have only some minor revision to add:

Please change the reference number 1 that is a link to a non governative web site and change it with GBD-2019 psoriasis related data published on Frontiers in Med.

Reply:

Thank you for this comment. We corrected the references:

Damiani, G.; Bragazzi, N.L.; Karimkhani Aksut, C.; Wu, D.; Alicandro, G.; McGonagle, D.; Guo, C.; Dellavalle, R.; Grada, A.; Wong, P.; et al. The Global, Regional, and National Burden of Psoriasis: Results and Insights From the Global Burden of Disease 2019 Study. Front Med (Lausanne) 2021, 8, 743180, doi:10.3389/fmed.2021.743180.

Please mention carefully the cardiovascular implications of inflammatio based low levels of vitamin D.

Please describe the cardiovascular comorbidities in psoriasis with a particular attention the endotypes [10.3390/jcm9010186, 10.1016/j.jid.2019.07.727]

Reply:

Thank you for this comment. We added the following information:

It should be noted that psoriatic patients are more likely to develop severe vascular events such as myocardial infarction and stroke (up to 50%) that general population [53,54]. In addition, among psoriaticsis patients, different endotypes with different risks, severity and treatment options are distinguished based on hypertension, red cell distribution width and mean platelet volume [55,56]

  1. Garshick, M.S.; Ward, N.L.; Krueger, J.G.; Berger, J.S. Cardiovascular Risk in Patients With Psoriasis: JACC Review Topic of the Week. J Am Coll Cardiol 2021, 77, 1670-1680, doi:10.1016/j.jacc.2021.02.009.
  2. Purzycka-Bohdan, D.; Kisielnicka, A.; Bohdan, M.; Szczerkowska-Dobosz, A.; Sobalska-Kwapis, M.; Nedoszytko, B.; Nowicki, R.J. Analysis of the Potential Genetic Links between Psoriasis and Cardiovascular Risk Factors. Int J Mol Sci 2021, 22, doi:10.3390/ijms22169063.
  3. Conic, R.R.; Damiani, G.; Schrom, K.P.; Ramser, A.E.; Zheng, C.; Xu, R.; McCormick, T.S.; Cooper, K.D. Psoriasis and Psoriatic Arthritis Cardiovascular Disease Endotypes Identified by Red Blood Cell Distribution Width and Mean Platelet Volume. J Clin Med 2020, 9, doi:10.3390/jcm9010186.
  4. Seth, D.; Ehlert, A.N.; Golden, J.B.; Damiani, G.; McCormick, T.S.; Cameron, M.J.; Cooper, K.D. Interaction of Resistin and Systolic Blood Pressure in Psoriasis Severity. J Invest Dermatol 2020, 140, 1279-1282 e1271, doi:10.1016/j.jid.2019.07.727.

Reviewer 3 Report

A very complete narrative review exploring the role of vitamin D in psoriasis, and also its involvement in disease treatment; minor revisions are requested before the paper may be considered eligible to be published:

Various typos are sparce throughout the paper, so an English revision is required; for example in the conclusions: "Thus, targeting vitamin D signaling would represent promising futhure in therapy or prevention of psoriais." please check.

line 508 you should add: " Different clinical studies report the combination of calcipotriene and betamethasone dipropionate in foam solution to be the most effective treatment for mild psoriasis" and cite: doi: 10.1111/dth.13185.

line 78 you should add: "for this reason, biologic drugs targeting IL such as IL 17, 23 and TNF alpha are the mainstay in the management of severe psoriasis" and cite: "doi: 10.3390/pharmaceutics14020294." and doi: 10.1111/dth.14504.

Good Luck!

Author Response

Reviewer 3

A very complete narrative review exploring the role of vitamin D in psoriasis, and also its involvement in disease treatment; minor revisions are requested before the paper may be considered eligible to be published:

Reply:

We thank the reviewer for his/her attention and positive comments.

Various typos are sparce throughout the paper, so an English revision is required; for example in the conclusions: "Thus, targeting vitamin D signaling would represent promising futhure in therapy or prevention of psoriais." please check.

Reply:

Thank you for this comment. We corrected the manuscript with Track changes option.

line 508 you should add: " Different clinical studies report the combination of calcipotriene and betamethasone dipropionate in foam solution to be the most effective treatment for mild psoriasis" and cite: doi: 10.1111/dth.13185.

line 78 you should add: "for this reason, biologic drugs targeting IL such as IL 17, 23 and TNF alpha are the mainstay in the management of severe psoriasis" and cite: "doi: 10.3390/pharmaceutics14020294." and doi: 10.1111/dth.14504.

Reply:

Thank you for this comment. We corrected the manuscript according to Reviewer’s suggestions.

Good Luck!

Reply:

Thank you
